# Analysis of Intestinal Metabolites in SR−B1 Knockout Mice via Ultra−Performance Liquid Chromatography Quadrupole Time−of−Flight Mass Spectrometry

**DOI:** 10.3390/molecules28020610

**Published:** 2023-01-06

**Authors:** Qijun Chen, Lixue Wang, Jinlong Chen, Hui Song, Wen Xing, Ziqian Wang, Xueying Song, Hua Yang, Wenhua Zhao

**Affiliations:** 1School of Pharmaceutical Sciences, Capital Medical University, Beijing 100069, China; 2School of Basic Medical Sciences, Capital Medical University, Beijing 100069, China; 3Central Laboratory, Capital Medical University, Beijing 100069, China

**Keywords:** scavenger receptor class B type 1, intestinal metabolites, lipid, amino acid, neurotransmitter

## Abstract

Scavenger receptor class B type 1 (SR−B1), a multiligand membrane receptor, is expressed in a gradient along the gastrocolic axis. SR−B1 deficiency enhances lymphocyte proliferation and elevates inflammatory cytokine production in macrophages. However, whether SR−B1 affects intestinal metabolites is unclear. In this study, we detected metabolite changes in the intestinal tissue of SR−B1^−/−^ mice, including amino acids and neurotransmitters, by ultra−performance liquid chromatography quadrupole time−of−flight mass spectrometry (UHPLC−Q−TOF/MS) and HPLC. We found that SR−B1^−/−^ mice exhibited changes in intestinal lipid metabolites and metabolic pathways, including the glycerophospholipid, sphingolipid, linoleic acid, taurine, and hypotaurine metabolic pathways. SR−B1 deficiency influenced the contents of amino acids and neurotransmitters in all parts of the intestine; the contents of leucine (LEU), phenylalanine (PHE), tryptophan (TRP), and tyrosine (TYR) were affected in all parts of the intestine; and the contents of 3,4−dihydroxyphenylacetic acid (DOPAC) and dopamine (DA) were significantly decreased in both the colon and rectum. In summary, SR−B1 deficiency regulated intestinal lipids, amino acids, and neurotransmitter metabolism in mice.

## 1. Introduction

Scavenger receptor class B type 1 (SR−B1), which belongs to the class B scavenger receptor family, is a high−density lipoprotein (HDL) receptor [1]. SR−B1, with a molecular weight of approximately 82 kDa, is a 509−amino−acid multiligand membrane glycoprotein with extracellular domains anchored to the plasma membrane at the N− and C−terminations by hydrophobic transmembrane regions that extend into the cytoplasm [2]. SR−B1 is predominantly expressed in the liver and steroidogenic tissues. SR−B1 is also highly expressed in immune−response−related phagocytic cells, which are represented by inflammation−related macrophages (Ms), dendritic cells (DCs), mature monocytes, and neutrophils [3]. Importantly, SR−B1 is expressed in the intestines, primarily on the apical surfaces of epithelial cells [4,5]. Their expression levels follow a gradient along the gastrocolic axis of the intestine, with the highest level of expression in the proximal intestine and decreasing to minimal expression levels in the distal intestine [6]. SR−B1 participates in cholesterol balance by selectively removing cholesteryl esters from HDLs, bidirectionally regulating unesterified cholesterol movement [7]. It was reported that the plasma total cholesterol in SR−B1 knockout mice was significantly increased [8,9], and the plasma cholesterol concentration of heterozygous and homozygous mutant mice increased by approximately 31% and 125%, respectively [10]. SR−B1 induces endothelial cell migration [11], activates endothelial nitric oxide synthase [12], inhibits Ms apoptosis [13], regulates lymphocyte autoimmunity and homeostasis [14], and initiates cell−signaling events [15]. Furthermore, the class B scavenger receptor mediates the internalization, adhesion, and phagocytosis of various bacteria [16]. SR−B1 is related to lipids and lipopolysaccharide (LPS) in Gram−negative bacteria, which may accelerate LPS participation in chylomicrons and lipoprotein transportation [17]. However, SR−B1’s effects on intestinal metabolite levels have not been reported. Therefore, in this study, we examined the intestinal metabolites in SR−B1^−/−^ mice.

## 2. Results

### 2.1. Detection of the Expression of SR−B1 mRNA by RT−PCR

The C57 mouse gene sequence was AGTCTCAGGCAGCTGTTGCAGAGCCGTAAAGTGGGGAAGCCCCTCCTCACATCCTCCCTGTGTCTCCC……CCCTGATAAATGTCCCACCGGCACGCGTTATCCACAAGCCGTGTCGCTCCGAAGTCCGAAGGGTCCTGCCCCGAGGTTAAGATTCCATCAGTGG. The SR−B1^−/−^ mouse gene sequence was AGTCTCAGGCAGCTGTTGCAGAGCCGTAA…(−10612bp)…CGCTCCGAAGTCCGAAGGGTCCTGCCCCGAGGTTAAGATTCCATCAGTGG (Figure 1A). As shown in Figure 1B, mice 1, 4, 5, and 6 had no bands in the P1 and P2 PCR, but they produced 678 bp bands in the P3 and P4 PCR, indicating they were wild−type (C57) mice. Mouse 3 produced 654 and 678 bp bands, indicating it was a heterozygous mutant mouse (SR−B1^+/−^). Mice 2 and 7 had no bands in the P3 and P4 PCR, but they produced 654 bp bands in the P1 and P2 PCR, indicating they were homozygous mutant mice (SR−B1^−/−^).

### 2.2. Scavenger Receptor Class B Type 1 Deficiency Influences Intestinal Metabolomics

The results of the PLS−DA and PCoA are shown in Figure 2. Samples of C57 and SR−B1^−/−^ mice were well separated in the positive−ion and negative−ion modes. The metabolomics of PLS−DA and PCoA showed that the metabolite compositions in the SR−B1^−/−^ mice were significantly different from those in the C57 mice, as determined by UHPLC-Q-TOF/MS in the positive−ion or negative−ion mode (Figure 2).

The contents of the differential metabolite abundance in the small intestine and large intestine of the SR−B1^−/−^ mice were generally increased, and most of the differentially abundant metabolites were lipids. Compared with the C57 mice, 10 metabolites in the small intestine of the SR−B1^−/−^ mice were significantly reduced, and 40 metabolites were significantly increased; 3 metabolites in the large intestine of the SR−B1^−/−^ mice were significantly reduced, and 33 metabolites were significantly increased (Figure 3A–D). For example, the cardiolipin (CL) (i−13:0/i−21:0/18:2(9Z,11Z)/i−24:0) and tauro−b−muricholic acid abundance was significantly increased, and the taurodeoxycholic acid abundance was significantly decreased in the SR−B1^−/−^ mouse small intestine compared to the C57 mouse small intestine. The phosphatidylethanolamine (PE−NMe) (18:1(11Z)/22:6(4Z,7Z,10Z, 13Z,16Z,19Z)), ganglioside GM1 (d18:1/26:0), ganglioside GM1 (d18:0/24:0), and phosphatidylcholine (PC) (22:6(4Z,7Z,10Z, 13Z,16Z,19Z)/16:1(9Z)) levels were significantly increased, while the lysoPC (P−18:1(9Z)) and PE−NMe (20:2(11Z,14Z)/18:0) levels were obviously decreased in the SR−B1^−/−^ large intestine.

The KEGG metabolic pathway analysis showed that glycerophospholipid metabolism and linoleic acid metabolism were obviously different in both the small intestine and large intestine in the SR−B1^−/−^ mice compared with those in the C57 mice (*p* < 0.05), while taurine and hypotaurine metabolism were obviously different in the small intestine (*p* < 0.05), and the sphingolipid metabolism was obviously different in the large intestine (*p* < 0.05, Figure 4A,B). The metabolic pathways of the glycerophospholipid metabolism and sphingolipid metabolism were closely related to glycine (GLY), threonine (THR), and serine (SER) metabolism, and the metabolism of taurine (TAU) and hypotaurine were closely related to alanine (ALA), TAU, and methionine (MET) metabolism. The metabolite enrichment analysis showed that glycerophosphoethanolamines, glycerophosphocholines, TAU conjugates, ceramide phosphocholines, and others were significantly enriched in the small intestine and that diacylglycerophosphoserines, glycosphingolipids, phosphosphingolipids, and others were significantly enriched in the large intestine (Figure 4C,D).

### 2.3. Scavenger Receptor Class B Type 1 Deficiency Influences the Amino Acid Contents in the Intestine

Compared with those in the C57 mice, the levels of LYS, leucine (LEU), isoleucine (ILE), PHE, valine (VAL), MET, tryptophan (TRP), γ−aminobutyric acid (GABA), TYR, THR, and arginine (ARG) in the duodenum of the SR−B1^−/−^ mice were significantly increased (*p* < 0.001), while the levels of TAU, GLY, SER, glutamic acid (GLU), and aspartic acid (ASP) were significantly decreased (Figure 5, *p* < 0.05). Compared with those in the C57 mice, the levels of LEU, GABA, ILE, PHE, TYR, ARG, THR, and TRP in the jejunum of the SR−B1^−/−^ mice were obviously elevated (*p* < 0.05), while the levels of ALA, TAU, SER, and ASP were significantly decreased (*p* < 0.05). Compared with those in the C57 mice, the LYS, LEU, ILE, PHE, MET, TRP, TYR, THR, and ARG levels in the ileum of the SR−B1^−/−^ mice were obviously increased (*p* < 0.01), while the levels of GABA, TAU, GLY, and SER were significantly decreased (*p* < 0.05). The levels of LYS, ILE, LEU, PHE, GABA, TRP, TYR, MET, ALA, ASP, THR, SER, ARG, and GLY in the colon of the SR−B1^−/−^ mice were obviously lower than those in the colon of the C57 mice (*p* < 0.05). Compared with those in the C57 mice, the levels of LYS, LEU, PHE, and TRP in the rectum of the SR−B1^−/−^ mice were obviously increased (*p* < 0.01), whereas the levels of GABA, TYR, ALA, GLY, and SER were significantly reduced (*p* < 0.05).

### 2.4. Scavenger Receptor Class B Type 1 Deficiency Influences the Neurotransmitter Contents in the Intestine

Compared with those in the C57 mice, the levels of 5−hydroxyindoleacetic acid (HIAA), 5−hydroxytryptophan (HTP), 5−hydroxytryptamine (5−HT), homovanillic acid (HVA), and 3,4−dihydroxyphenylacetic acid (DOPAC) in the duodenum of the SR−B1^−/−^ mice were significantly increased (*p* < 0.01). Compared with those in the C57 mice, the HIAA, HTP, 5−HT, and DOPAC levels in the jejunum of the SR−B1^−/−^ mice were obviously increased (*p* < 0.05), and the HIAA and HTP levels in the ileum of the SR−B1^−/−^ mice were significantly increased (*p* < 0.05). The HIAA, DOPAC, and DA levels in the colon of the SR−B1^−/−^ mice were significantly reduced (*p* < 0.05), while the level of HVA was significantly increased (*p* < 0.05). The levels of adrenaline (E), 5−HT, and HVA in the rectum of the SR−B1^−/−^ mice were obviously increased (*p* < 0.05), while the levels of DOPAC and dopamine (DA) were obviously decreased (*p* < 0.001, Figure 6).

## 3. Discussion

In the present work, we found that SR−B1 knockout affected the intestinal metabolites of mice. The SR−B1^−/−^ mice showed regulated intestinal glycerophospholipid, linoleic acid, sphingolipid, taurine, and hypotaurine metabolism and altered amino acid and neurotransmitter contents.

Glycerophospholipids regulate inflammation, immunity, and tumor development [18,19] and were shown to be effective guardians of intestinal epithelial homeostasis [20,21]. Glycerophospholipid metabolism plays a key pathogenic role in the occurrence and progression of atherosclerosis [22]. This study found that there were significant changes in the levels of phosphatidylcholine (PC) and lysophosphatidylcholine (LysoPC) in the intestines of SR−B1 knockout mice. PC, a subclass of glycerophospholipids found in all major lipoproteins, is not only a major component of biofilms but also acts as a signaling molecule and bioactive mediator in atherosclerosis−associated cellular processes such as apoptosis, proliferation, and inflammation [23,24]. LysoPC is the main component of oxidized LDL and has a variety of biological functions in cardiovascular disease, and LysoPC and sphingolipids can be used as biomarkers for atherosclerosis [25]. A study found that LysoPC levels in atherosclerosis plaques were significantly correlated with IL−1β, interleukin−6, tumor necrosis factor−α, oxidative stress, and chemoattotic proteins [26]. Sphingolipids play an important role in the cardiovascular system [27], and in sphingolipid metabolism, ceramides are key compounds formed by sphingolipids through sphingomyelinase [28]. Activation of the sphingomyelinase–ceramide pathway promotes proinflammatory and pro−oxidative activity (e.g., through oxidized LDL) and mediates calcification of vascular smooth muscle cells, leading to atherosclerosis and other cardiovascular diseases [29]. *Bacteroides* can produce sphingolipids, and microbially derived sphingolipid deficiency has been related to IBD [30]. Linoleic acid lowers blood cholesterol and prevents atherosclerosis [31,32]. Studies have found that cholesterol must be combined with linoleic acid in order to function and metabolize properly in the body.

In addition, taurine and hypotaurine metabolism have been related to gastrointestinal injury [33]. In glycerophospholipid metabolism, PHE, TYR, and TRP biosynthesis and PHE metabolism have been reported to regulate intestinal inflammation and oxidative stress [34]. It was reported that some amino acids can be metabolized into neurotransmitters; for example, TRY was catalyzed by tryptophan hydroxylase to generate HTP, catalyzed by 5−hydroxytryptophan decarboxylase to 5−HT, and then metabolized to 5−HIAA [35,36]. TYR was converted to levodopa by tyrosine hydroxylase, which was then converted to DA by dopa decarboxylase, and then metabolized to DOPAC and HVA [37]. Furthermore, the gut microbiota can directly synthesize neurotransmitters such as 5−HT, NE, and DA, and changes in intestinal 5−HT levels have been associated with mucosal inflammation in colitis [38,39].

SR−B1 selectively takes up cholesterol esters from HDL and exhibits recognized antiatherogenic reverse cholesterol transport ability [40,41]. SR−B1 was used as a target for therapeutic drugs delivered through recombinant HDL [42], and SR−B1 influences the sensitivity to LPS−induced inflammation [40,43]. SR−B1 deficiency thus plays a key role in promoting the acquisition of a proinflammatory phenotype. Furthermore, SR−B1 is highly expressed in cancers [42]. Accordingly, the HDL receptor SR−B1 appears to be involved in oncogenic metabolism. Moreover, emerging evidence indicates that SR−B1 functions in other cellular processes, including the induction of endothelial cell migration [11], inhibition of Ms apoptosis [13], and regulation of lymphocyte autoimmunity and homeostasis [14]. In humans, intestinal inflammatory disorders may be due to intestinal metabolite and microbiota dysbiosis. Intestinal metabolites, the microbiota, and immune status significantly impact intestinal diseases by providing nutrients and defending against pathogen invasion. Many diseases are related to immune dysbiosis caused by intestinal metabolite and microbiota dysbiosis, such as IBD, colorectal cancer, and immune disorders [40,44,45]. Our present study shows that SR−B1 knockout regulated intestinal metabolites. Further research to determine whether SR−B1 knockout affects the development of gut−related diseases such as colitis and colorectal cancer is urgently needed.

## 4. Materials and Methods

### 4.1. Animals

A total of 10 specific−pathogen−free healthy male 8−week−old C57BL/6 mice were purchased from Vital River Laboratories (Beijing Vital River Laboratory Animal Technology Co., Ltd., Beijing, China). Using CRISPR/Cas9 technology, nonhomologous recombinant repair was used to introduce mutations to obtain SR−B1 gene knockout mice. The SR−B1^−/−^ mice were purchased from Shanghai Model Organisms Center, Inc. (Shanghai, China), and the genotypes of 10 male 8−week−old SR−B1^−/−^ or SR−B1^+/−^ mice were characterized through polymerase chain reaction (PCR). The animals were housed in standard−sized cages at room temperature, 24 ± 1 °C, with 60 ± 5% humidity, in a 12 h light/dark cycle. The animals were given plenty of regular feed and purified water. The Experimental Animal Management Committee of the Capital Medical University approved all studies (IACUC protocol no.: AEEI−2019−070). All the experiments were performed following the guidelines of the Experimental Animal Care and Use Committee of Capital Medical University (Beijing, China).

### 4.2. Detection of SR−B1 mRNA Expression by RT–PCR

Mouse tail DNA was extracted using a Universal Genomic DNA Purification Mini Spin Kit (Beyotime, D0063). Then, DNA was amplified with the primers 5′AATGGACCCTGTGCTTGGAGTG3′ (P1), 5′GGAGGAGGAGGTGGTCATAG AACG3′ (P2), 5′TCCTAATCCTTCCAAGCCGTTCTC3′ (P3), and 5′CAGCCATTTTGCCCATTT TGTGC3′ (P4). The PCR amplification program was set for predenaturation at 95 °C for 5 min, and then denaturation for 15 s at 95 °C, annealing at 58 °C for 15 s, and extension for 60 s at 72 °C in 35 cycles. The PCR products were electrophoresed in a 1% agarose gel and visualized under ultraviolet (UV) light at a wavelength of 300 nm. 

### 4.3. Metabolomics Analysis of the Intestines

The small and large intestine tissues were weighed and homogenized, vortexed for 3 min, and dissolved in an 80% methanol solution 10 times. The samples were then vortexed for 1 min, sonicated for 10 min, maintained at −20 °C for 30 min, and centrifuged at 18,000× *g*/min for 20 min at 4 °C. The supernatant was poured into an ampoule bottle, frozen at −80 °C, and placed in a lyophilizer. Then, 1.5 mL of a 50% methanol solution was added to the lyophilized sample powder to reconstitute the powder with sonication for 10 s. Then, 0.5 mL of acetonitrile was added to 0.5 mL of the reconstituted solution and subjected to sonication to promote dissolution. The samples were then centrifuged at 18,000× *g*/min for 10 min at 4 °C, and the supernatants were poured into a vial for testing. Specifically, supernatants were collected for analysis by ultra−performance liquid chromatography quadrupole time−of−flight mass spectrometry (UHPLC−Q−TOF/MS). At the same time that a test sample was prepared, a bulk quality control sample was prepared by mixing an equal volume of each sample. The quality control samples were used to identify the LC–MS peaks.

### 4.4. Determination of Amino Acid Contents in the Intestines

HPLC with fluorescence detection (FLD) was performed to determine the amino acid contents in the intestines. The sample preparation method was consistent with the sample preparation method for detecting neurotransmitters. A ZORBAX Eclipse XDB−C8 chromatography column (4.6 × 150 mm, 5 μm) was maintained at 30 °C, and the flow rate was 1 mL/min. The mobile phases consisted of methanol (A) and 60 mmol/L sodium citrate (B, pH = 6), and the gradient elution procedure is shown in Table 1. The excitation wavelength was 335 nm, and the emission wavelength was 460 nm. The HPLC instrument automatically injected the sample by combining 2.0 μL of the standard or sample and 8.0 μL of derivatization reagent (0.04 mol/L o−phthalaldehyde, 2.5 mol/L methanol, 0.06 mol/L 2−hydroxy−1−ethanethiol, and 0.1 mol/L borax buffer at pH = 9.8). Then, 10 microliters of the sample mix solution was mixed 6 times at maximum speed while exposed to air and allowed to stand for 1 min before injection.

### 4.5. Determination of Neurotransmitter Contents in the Intestines

The intestine from each mouse was weighed and quickly homogenized in 120 L of pretreatment solution A (0.4 mol/L perchloric acid) on ice and then centrifuged at 12,000 rpm/min for 20 min at 4 °C after standing at room temperature for 30 min. Then, 90 μL of supernatant was collected, and 45 μL of pretreatment solution B (20 mmol/L of potassium citrate, 0.3 mol/L of dipotassium hydrogen phosphate, and 2 mmol/L of EDTA·2Na) was added. The sample was vortexed and then centrifuged at 12,000 rpm/min for 20 min at 4 °C after standing at room temperature for 30 min. The supernatants were collected and analyzed by HPLC coupled with an electrochemical detector (Waters ECD2465, Milford, MA, USA). An XBridge^TM^ amide chromatography column (150 × 4.6 mm, 5 μm) was maintained at 30 °C, and the flow rate was 0.8 mL/min. The mobile phase consisted of buffer salt solution (50 mmol/L citric acid monohydrate, 70 mmol/L sodium acetate anhydrous, 10 mmol/L EDTA·2Na, and 180 mmol/L 1−octanesulfonate)–methanol (92:8, *V*/*V*). The detector potential was +0.6 V, with an injection volume of 10 μL.

### 4.6. Statistical Analysis

The metabolomics data were preprocessed with Progenesis QI v2.3 software (Waters, Elstree, UK). The compound characterization was performed using the Human Metabolome Database (HMDB), LIPID MAPS (v2.3), and the METLIN database, and the metabolic pathway analyses were performed using the Kyoto Encyclopedia of Genes and Genomes (KEGG) database. The GraphPad Prism 8.0.1 (GraphPad Software, California, CA, USA), and R software packages (TUNA Team, Tsinghua University, Beijing, China) were used for the statistical analysis. The data are expressed as the mean ± standard deviation (SD). One−way ANOVA was performed to compare multiple groups.

## Figures and Tables

**Figure 1 molecules-28-00610-f001:**
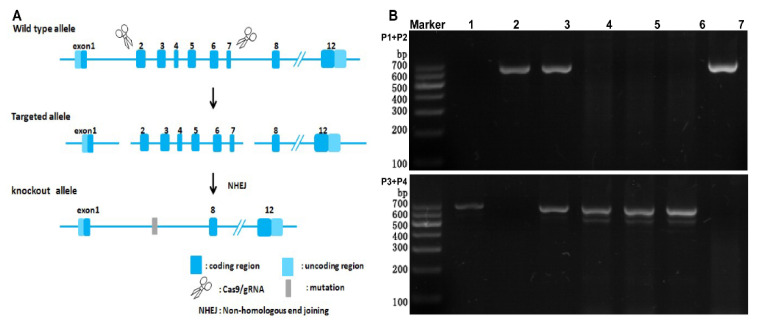
Strategy for the targeted disruption of the SR−B1 locus in mice (**A**); detection of the expression of SR−B1 mRNA by RT−PCR, 1, 4, 5, and 6—C57 mice, 3—SR−B1^+/−^ mouse, 2 and 7—SR−B1^−/−^ mice (**B**).

**Figure 2 molecules-28-00610-f002:**
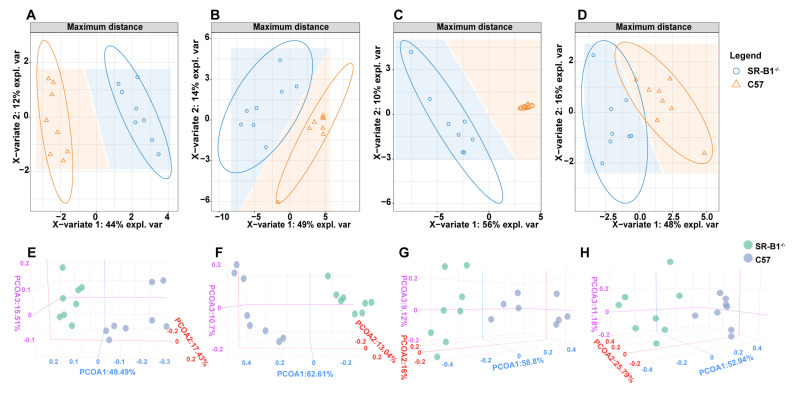
Partial least squares discriminant analysis (PLS−DA) and principal coordinate analysis (PCoA) of the small intestine in the C57 and SR−B1^−/−^ mice in the negative−ion (**A**,**E**) or positive−ion (**B**,**F**) mode (*n* = 8); PLS−DA and PCoA of the large intestine in the C57 and SR−B1^−/−^ mice in the negative−ion (**C**,**G**) or positive−ion (**D**,**H**) mode.

**Figure 3 molecules-28-00610-f003:**
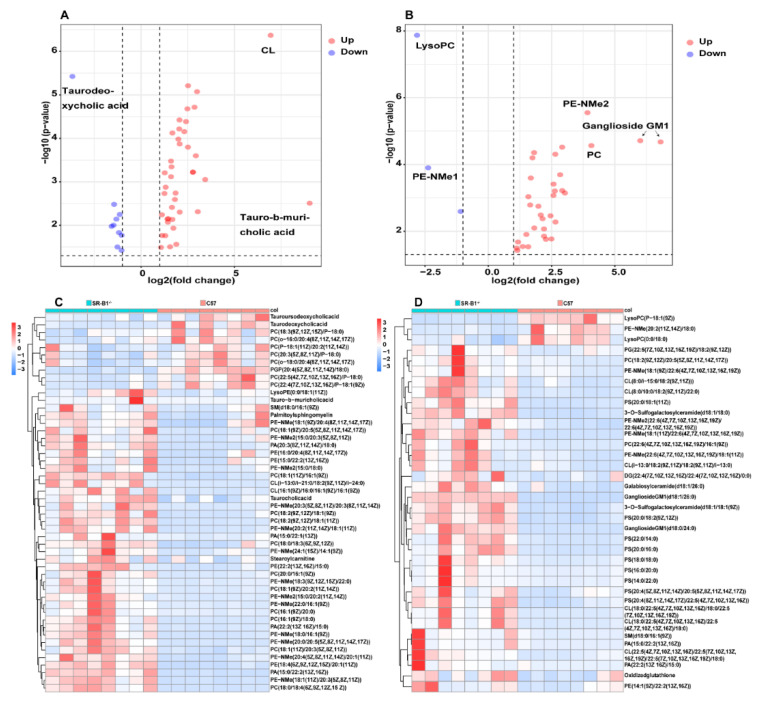
The differential metabolites in the small intestine (**A**,**C**) and large intestine (**B**,**D**) of the C57 and SR−B1^−/−^ mice; *p* < 0.05, fold change >2, and variable importance in projection (VIP) >1.

**Figure 4 molecules-28-00610-f004:**
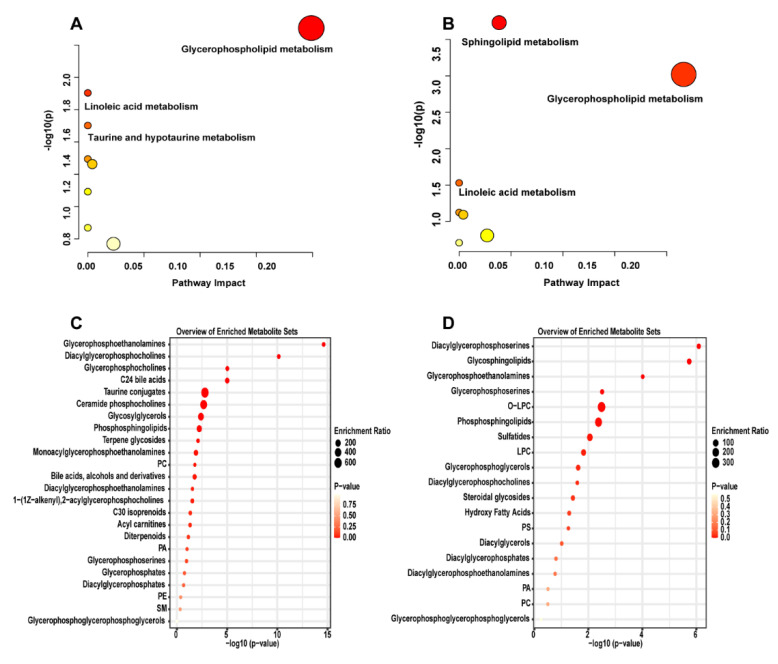
KEGG pathway analysis of differential metabolites in the small intestine (**A**) and large intestine (**B**) of the C57 and SR−B1^−/−^ mice. Metabolite enrichment analysis of the small intestine (**C**) and large intestine (**D**) in the C57 and SR−B1^−/−^ mice.

**Figure 5 molecules-28-00610-f005:**
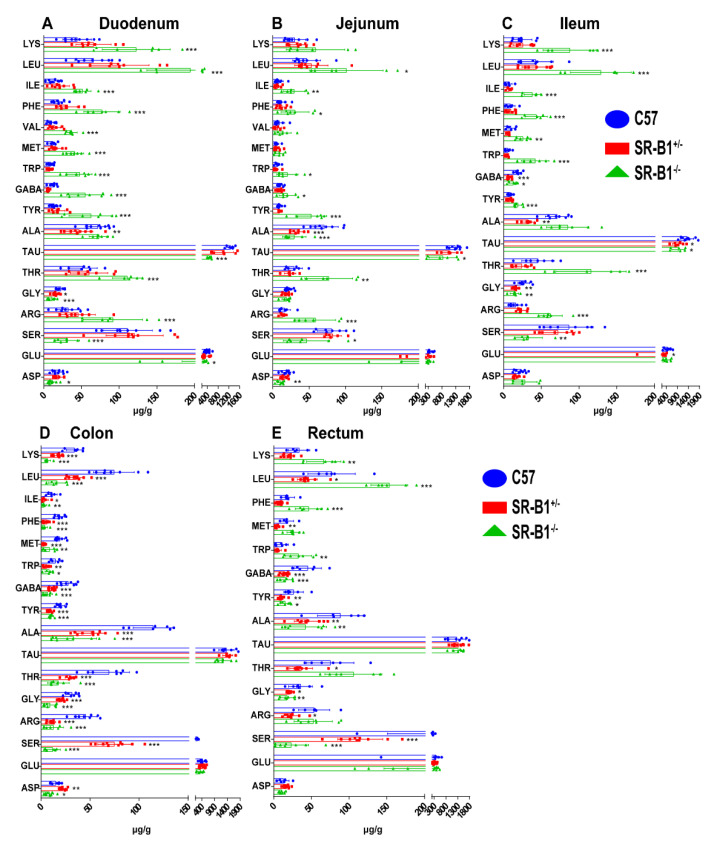
The difference in amino acid contents among SR−B1^−/−^, SR−B1^+/−^, and C57 mice (*n* = 6–10). Determination of the amino acid contents in the duodenum (**A**), jejunum (**B**), ileum (**C**), colon (**D**), and rectum (**E**) of the C57, SR−B1^+/−^, and SR−B1^−/−^ mice. The data are presented as the means ± SD; * *p* < 0.05, ** *p* < 0.01, and *** *p* < 0.001 compared with the C57 mice. LYS (lysine), LEU (leucine), ILE (l−isoleucine), PHE (phenylalanine), VAL (valine), MET (DL−methionine), TRP (tryptophan), GABA (γ−aminobutyric acid), TYR (tyrosine), ALA (alanine), TAU (taurine), THR (L−threonine), GLY (glycine), ARG (arginine), SER (serine), GLU (glutamic acid), and ASP (aspartic acid).

**Figure 6 molecules-28-00610-f006:**
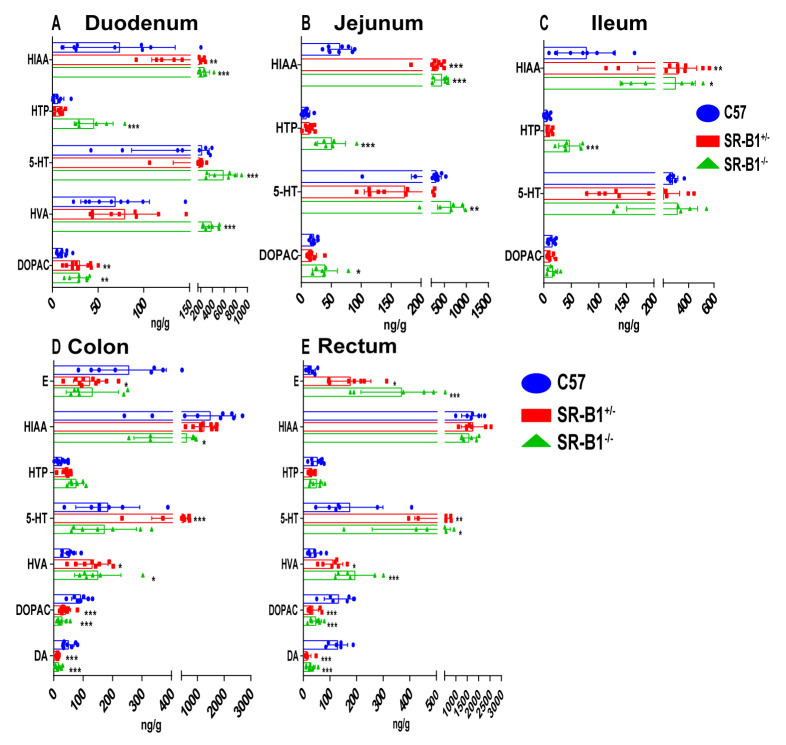
The difference in neurotransmitter contents among SR−B1^−/−^, SR−B1^+/−^, and C57 mice (*n* = 6–10). Determination of neurotransmitter contents in the duodenum (**A**), jejunum (**B**), ileum (**C**), colon (**D**), and rectum (**E**) of the C57, SR−B1^+/−^, and SR−B1^−/−^ mice. The data are presented as the means ± SD; * *p* < 0.05, ** *p* < 0.01, and *** *p* < 0.001 compared with the C57 mice. E (adrenaline), HIAA (5−hydroxyindoleacetic acid), HTP (5−hydroxytryptophan), 5−HT (5−hydroxytryptamine), HVA (homovanillic acid), DOPAC (3,4−dihydroxyphenylacetic acid), and DA (dopamine).

**Table 1 molecules-28-00610-t001:** The gradient elution procedure of the HPLC.

Time (min)	A (%)	B (%)
0.0	21.0	79.0
18.0	21.0	79.0
19.0	28.0	72.0
40.0	28.0	72.0
41.0	32.0	68.0
44.0	32.0	68.0
46.0	36.0	64.0
47.0	46.5	53.5
52.0	46.5	53.5
53.0	49.5	50.5
58.0	49.5	50.5
59.0	59.0	41.0
64.0	59.0	41.0
64.5	70.0	30.0
70.0	70.0	30.0

## Data Availability

All data generated or analyzed during this study are included in this published article.

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
