# Peer review of "Analysis of Intestinal Metabolites in SR−B1 Knockout Mice via Ultra−Performance Liquid Chromatography Quadrupole Time−of−Flight Mass Spectrometry"

_molecules, 2023, doi:10.3390/molecules28020610_

Round 1

Reviewer 1 Report

The author's paper appears to be a new result. But you need four supplement them.

1) Metabolism has a lot to do with age. The authors did not indicate the mouse age. If you used a young mouse or an old mouse, please writing  the reason.

2) SR-B1 gene knockout mice lack genetic information. In Figure 1, simple mRNA expression information is unclear. Therefore, genetic maps(site a gene knockout involved both exon and intron),  and primers position notation should be adding in Figure1 relate a reference (Biomedisines: 2021, 9, 612. https://doi.org/10.3390/biomedicines9060612,  See Figure 2A).

3) Please present knockout mice evidence. For example, increased HDLc or total cholesterol in SR-B1 knockout mice

4) Opinions on factors that increase various lipids in the intestine or decrease amino acids and neurotransmitters are required for consideration. Also, please be the discussion of the association of the authors' findings with factors that increase atherosclerosis and inflammation in SR-B1 knockout mice.

Author Response

Dear Editors and Reviewers,

Thank you very much for your valuable comments and suggestions on the paper. We have carefully revised the article based on your comments, and marked the changes in red font. The answers to each question are as follows:

Reviewer 1

Q1: Metabolism has a lot to do with age. The authors did not indicate the mouse age. If you used a young mouse or an old mouse, please writing the reason.

Reply: We selected 8 weeks mice in their prime for metabolomic testing, when all organs of the mice were mature, the life and health index was good, and all aspects of the mice were in a very healthy state. After 6 months, mice will enter the middle and old age, the elderly mice often have weak immunity, slow response, reduced function, liver and kidney function gradually fail, and old mice often have various diseases. The old mice have many interference factors, which are not conducive to obtaining accurate experimental data. We made modifications in lines 235-243 as shown below.

Animals

A total of 10 specific-pathogen-free healthy male 8 weeks C57BL/6 mice were purchased from Vital River Laboratories (Beijing Vital River Laboratory Animal Technology Co., Ltd., Beijing, China). Using CRISPR/Cas9 technology, non-homologous recombinant repair was used to introduce mutations, resulting in SR-B1 gene protein code reading frame code shifting, loss of function, and obtaining SR-B1 gene knockout mice. SR-B1+/- mice were purchased from Shanghai Model Organisms Center, Inc. (Shanghai, China), and the genotypes of 10 male 8 weeks SR-B1-/- or SR-B1+/- mice were characterized through polymerase chain reaction (PCR).

Q2: SR-B1 gene knockout mice lack genetic information. In Figure 1, simple mRNA expression information is unclear. Therefore, genetic maps (site a gene knockout involved both exon and intron), and primers position notation should be adding in Figure1 relate a reference (Biomedisines: 2021, 9, 612. https://doi.org/10.3390/biomedicines9060612, See Figure 2A).

Reply: We have added genetic information and primer position information as shown below (lines 56-72).

Fig1 Strategy for targeted disruption of the SR-B1 locus in the mouse (A); Detection of the expression of SR-B1 mRNA by RT-PCR (B, 1, 4, 5, 6 were C57 mice, 3 was SR-B1+/- mouse, 2, 7 were SR-B1-/- mice)

The C57 mouse gene sequence was AGTCTCAGGCAGCTGTTGCAGAG CCGTAAAGT GGGGAAGCCCCTCCTCACACTCCTCCCTGTGTCTCCC……CCCTGATAAATGTCCCACCGGCACGCGTTATCCACAAGCCGTGTCGCTCCGAAGTCCGAAGGGTCCTGCCCCGAGGTTAAGATTCCATCAGTG. The SR-B1-/-mouse gene sequence was AGTCTCAGGCAGCTGT TGCAGAGCCGTAA…(-10612bp)…CGCTCCGAAGTCCGAAGGGTCCTGCCCCGAGGTTA AGATTCCATCAGTGG. As shown in Fig 1 B, the mice of 1, 4, 5, 6 had no bands in P1 and P2 PCR, they produced 678 bp bands in P3 and P4 PCR, which were wild-type (C57) mice. The mice of 3 produced 654 and 678 bp bands, which was heterozygous mutant mouse (SR-B1+/-). The mice of 2, 7 had no bands in P3 and P4 PCR, they produced 654 bp bands in P3 and P4 PCR, which were homozygous mutant mice (SR-B1-/-).

Q3: Please present knockout mice evidence. For example, increased HDLc or total cholesterol in SR-B1 knockout mice.

Reply:The classical effect of SR-B1 is to mediate the selective uptake of in vivo and in vitro HDL cholesterol esters through reverse cholesterol transport. We added the following to lines 43-46 in the study.

It was reported that the plasma total cholesterol in SR-B1 knockout mice were significantly increased [8,9], and the plasma cholesterol concentration of heterozygous and homozygous mutant mice increased by approximately 31% and 125%, respectively [10].

Q4: Opinions on factors that increase various lipids in the intestine or decrease amino acids and neurotransmitters are required for consideration. Also, please be the discussion of the association of the authors' findings with factors that increase atherosclerosis and inflammation in SR-B1 knockout mice.

Reply: We have revisited differential lipid metabolites in the intestine and discussed the association of our findings with factors of atherosclerosis and inflammation in SR-B1 knockout mice. We have modified lines 180-205 in the text as follows.

Glycerophospholipids regulate inflammation, immunity and tumour development [18,19], which was shown to be effective guardians of intestinal epithelium homeostasis [20]. Glycerophospholipid metabolism plays a key pathogenic role in the occurrence and progression of atherosclerosis [21]. This study found that in the levels of phosphatidylcholine (PC) and lysophosphatidylcholine (LysoPC) in the intestines of SR-B1 knockout mice were significant changes. PC, a subclass of glycerophospholipids, found in all major lipoproteins, is not only a major component of biofilms, but also acts as a signaling molecule and bioactive mediator in atherosclerosis-associated cellular processes such as apoptosis, proliferation, and inflammation [22,23]. LysoPC is the main component of oxidized LDL and has a variety of biological functions in cardiovascular disease, and LysoPC and sphingolipids can be used as biomarkers for atherosclerosis [24]. Study found that LysoPC levels in atherosclerosis plaques were significantly correlated with IL-1β, interleukin-6, tumor necrosis factor-α, oxidative stress and chemoattotic proteins [25]. Sphingolipids play an important role in the cardiovascular system [26], and in sphingolipid metabolism, ceramides are key compounds formed by sphingolipids through sphingomyelinase [27]. Activation of the sphingomyelinase-ceramide pathway promotes pro-inflammatory and pro-oxidative activity (e.g., through oxidized LDL), mediates calcification of vascular smooth muscle cells, leading to atherosclerosis and other cardiovascular diseases [28]. Bacteroides can produce sphingolipids, and microbially derived sphingolipid deficiency has been related to IBD [29]. Linoleic acid lowers blood cholesterol and prevents atherosclerosis [30, 31]. Studies have found that cholesterol must be combined with linoleic acid in order to function and metabolize properly in the body. If linoleic acid is lacking, cholesterol will combine with some saturated fatty acids, metabolic disorders occur, deposited on the blood vessel wall, and gradually form atherosclerosis, causing cardiovascular and cerebrovascular diseases.

Reviewer 2 Report

The authors investigate the metabolomic changes of murine intestine in case of SR-B1 knock-out. While missing this receptor will undoubtedly alter the lipid composition of many tissues, its main roles are in the liver and cardio-vascular system as it was shown that SR-B1 KO does not modify cholesterol absorption in the intestine. Still, the question proposed by the authors is valid. 

Issues:

The description of RT-PCR confirmation of SR-B1 KO (lines 59-63) uses mice always in plural. How many animals were analyzed total and how many per band ? 

Fig 2. is hardly visible, total number of samples included seems to be 8-8 per group but this is not described. It does not add much to the results, as it only shows that there are differences. PLS-DA or PCoA could be used for model building when a later classification of  samples is needed, which was not included in the current sample set. The only conclusion the authors draw from this is that the groups are different. For showing the differences the next figure, Fig. 3 is much better. In this figure one can see the distribution of values as well as the differences. Again only 8 samples. Where are the other two animals that they started with?

Figure 5 contains too many details, one can not even tell which color is what as the legend is impossible to read.

Figure 6, again legend is hard to read.

Discussion is in big extent repeating the introduction (sometimes copy-paste). The Kegg pathway analysis results are not discussed just some random statements (not related to KO effect or sometimes intestine metabolism) are listed for the metabolites which showed changed levels. Intestinal aminoacid metabolism is mixed with neuronal metabolism. A lot of the text deals with what is true when SR-B1 is present (eg. tumors).

Methods section does not describe the animal feeding, which is important if you analyze intestine. When were they last fed? With what? 

What happened with the intestinal content? was is scrapped away? 

Aminoacid composition in the methods section does not describe what was the sample for this measurement. Was it just the extracted tissue or was the tissue broken down somehow?

Author Response

Dear Editors and Reviewers,

Thank you very much for your valuable comments and suggestions on the paper. We have carefully revised the article based on your comments, and marked the changes in red font. The answers to each question are as follows:

Reviewer 2:

Q1: The description of RT-PCR confirmation of SR-B1 KO (lines 59-63) uses mice always in plural. How many animals were analyzed total and how many per band?

Reply: We apologize for the trouble caused by our incorrect description, and we have modified it in lines 67-72 as shown below. The genotypes of 10 male 8 weeks SR-B1-/- or SR-B1+/- mice were characterized, only 7 mice genotype identification results are shown in Figure 1.

As shown in Fig 1, the mice of 1, 4, 5, 6 had no bands in P1 and P2 PCR, they produced 678 bp bands in P3 and P4 PCR, which were wild-type (C57) mice. The mice of 3 produced 654 and 678 bp bands, which was heterozygous mutant mouse (SR-B1+/-). The mice of 2, 7 had no bands in P3 and P4 PCR, they produced 654 bp bands in P3 and P4 PCR, which were homozygous mutant mice (SR-B1-/-).

Q2: Fig 2. is hardly visible, total number of samples included seems to be 8-8 per group but this is not described. It does not add much to the results, as it only shows that there are differences. PLS-DA or PCoA could be used for model building when a later classification of samples is needed, which was not included in the current sample set. The only conclusion the authors draw from this is that the groups are different. For showing the differences the next figure, Fig. 3 is much better. In this figure one can see the distribution of values as well as the differences. Again only 8 samples. Where are the other two animals that they started with?

Reply: We have re-edited Figure 2 to make it visible. The visualization of PLS-DA and PCoA analysis showed significant differences in intestinal metabolite composition between SR-B1-/- and C57 mice, providing a basis for further analysis of differential metabolites. In the metabolomics experiment, 10 mice were detected in each group, but the samples of 8 mice in each group passed the quality inspection, so only the experimental results of 8 mice in each group were displayed. The total number of samples per group of 8-8 has been indicated in the Figure 2 legend, as shown below (lines 74-80).

Fig 2 Partial least squares discriminant analysis (PLS-DA) and principal coordinate analysis (PCoA) of the small intestine in the C57 and SR-B1-/-mice in negative-ion (A, E) or positive-ion (B, F) mode (n=8); PLS-DA and PCoA of the large intestine in the C57 and SR-B1-/- mice in negative-ion (C, G) or positive-ion (D, H) modes.

Q3: Figure 5 contains too many details, one can not even tell which color is what as the legend is impossible to read. Figure 6, again legend is hard to read.

Reply: We have re-edited Figures 5 and Figure 6 to make them clearer and easier to read, as shown below.

Fig 5 The difference in amino acid contents among SR-B1-/-, SR-B1+/- and C57 mice (n=6-10). Determination of amino acid contents in the duodenum (A), jejunum (B), ileum (C), colon (D) and rectum (E) of the C57, SR-B1+/- and SR-B1-/- mice. The data are presented as the means ± SD; * P<0.05, ** P<0.01 and *** P<0.001 compared with the C57 mice. LYS (lysine), LEU (leucine), ILE (l-isoleucine), PHE (phenylalanine), VAL (valine), MET (DL-methionine), TRP (tryptophan), GABA (γ-aminobutyric acid), TYR (tyrosine), ALA (alanine), TAU (taurine), THR (L-threonine), GLY (glycine), ARG (arginine), SER (serine), GLU (glutamic acid) ASP (aspartic acid)

Fig 6 The difference in neurotransmitter contents among SR-B1-/-, SR-B1+/- and C57 mice (n=6-10). Determination of neurotransmitter contents in the duodenum (A), jejunum (B), ileum (C), colon (D) and rectum (E) of the C57, SR-B1+/-and SR-B1-/- mice; The data are presented as the means ± SD; * P<0.05, ** P<0.01 and *** P<0.001 compared with the C57 mice. E (adrenaline), HIAA (5-hydroxyindoleacetic acid), HTP (5-hydroxytryptophan), 5-HT (5-hydroxytryptamine), HVA (homovanillic acid), DOPAC (3,4-dihydroxyphenylacetic acid), DA (dopamine)

Q4: Discussion is in big extent repeating the introduction (sometimes copy-paste). The Kegg pathway analysis results are not discussed just some random statements (not related to KO effect or sometimes intestine metabolism) are listed for the metabolites which showed changed levels. Intestinal aminoacid metabolism is mixed with neuronal metabolism. A lot of the text deals with what is true when SR-B1 is present (eg. tumors).

Reply: We have revised the discussion, removed some duplication, and focused on the KEGG pathway analysis results, as shown below (lines 180-205).

Glycerophospholipids regulate inflammation, immunity and tumour development [18,19], which was shown to be effective guardians of intestinal epithelium homeostasis [20]. Glycerophospholipid metabolism plays a key pathogenic role in the occurrence and progression of atherosclerosis [21]. This study found that in the levels of phosphatidylcholine (PC) and lysophosphatidylcholine (LysoPC) in the intestines of SR-B1 knockout mice were significant changes. PC, a subclass of glycerophospholipids, found in all major lipoproteins, is not only a major component of biofilms, but also acts as a signaling molecule and bioactive mediator in atherosclerosis-associated cellular processes such as apoptosis, proliferation, and inflammation [22,23]. LysoPC is the main component of oxidized LDL and has a variety of biological functions in cardiovascular disease, and LysoPC and sphingolipids can be used as biomarkers for atherosclerosis [24]. Study found that LysoPC levels in atherosclerosis plaques were significantly correlated with IL-1β, interleukin-6, tumor necrosis factor-α, oxidative stress and chemoattotic proteins [25]. Sphingolipids play an important role in the cardiovascular system [26], and in sphingolipid metabolism, ceramides are key compounds formed by sphingolipids through sphingomyelinase [27]. Activation of the sphingomyelinase-ceramide pathway promotes pro-inflammatory and pro-oxidative activity (e.g., through oxidized LDL), mediates calcification of vascular smooth muscle cells, leading to atherosclerosis and other cardiovascular diseases [28]. Bacteroides can produce sphingolipids, and microbially derived sphingolipid deficiency has been related to IBD [29]. Linoleic acid lowers blood cholesterol and prevents atherosclerosis [30, 31]. Studies have found that cholesterol must be combined with linoleic acid in order to function and metabolize properly in the body. If linoleic acid is lacking, cholesterol will combine with some saturated fatty acids, metabolic disorders occur, deposited on the blood vessel wall, and gradually form atherosclerosis, causing cardiovascular and cerebrovascular diseases.

Q5: Methods section does not describe the animal feeding, which is important if you analyze intestine. When were they last fed? With what? What happened with the intestinal content? was is scrapped away?

Reply: We have modified the animal feeding method (as shown below, lines 244-245), where the animals have been kept in an SPF level environment with regular feed and purified water, fasting the night before execution and given only purified water. The intestinal contents have been collected in -80 ℃ refrigerators.

The animals were housed in standard-sized cages at room temperature, 24 ± 1 °C, with 60% ± 5% humidity, in a 12-h light/dark cycle. The animals were given plenty of regular feed and purified water.

Q6: Amino acid composition in the methods section does not describe what was the sample for this measurement. Was it just the extracted tissue or was the tissue broken down somehow?

Reply: Amino acid content were measured in intestinal samples, and the sample preparation method was consistent with the sample preparation method for detecting neurotransmitters. We've made changes in this article as follows (lines 290-291).

HPLC with fluorescence detection (FLD) was performed to determine the amino acid contents in the intestines. The sample preparation method is consistent with the sample preparation method for detecting neurotransmitters.

Thank you again for your review of the manuscript.

References:

  1. Quiroz, A.; Molina, P.; Santander, N.; Gallardo, D.; Rigotti, A.; Busso, D. Ovarian cholesterol efflux: ATP-binding cassette transporters and follicular fluid HDL regulate cholesterol content in mouse oocytes.Biol Reprod2020, 102(2):348-361.
  2. Van Eck, M.; Twisk, J.; Hoekstra, M.; Van Rij, B.T.; Van der Lans, C.A.; Bos, I.S.; Kruijt, J.K.; Kuipers, F.; Van Berkel, T.J. Differential effects of scavenger receptor BI deficiency on lipid metabolism in cells of the arterial wall and in the liver. J Biol Chem2003, 278(26):23699-705.
  3. Rigotti, A.; Trigatti, B.L.; Penman, M.; Rayburn, H.; Herz, J.; Krieger, M. A targeted mutation in the murine gene encoding the high density lipoprotein (HDL) receptor scavenger receptor class B type I reveals its key role in HDL metabolism. Proc Natl Acad Sci USA1997, 94(23):12610-5.
  4. Makide, K.; Uwamizu, A.; Shinjo, Y.; Ishiguro, J.; Okutani, M.; Inoue, A.; Aoki, J. Novel lysophosphoplipid receptors: their structure and function.J Lipid Res2014, 55(10):1986-95.
  5. Wang, S.; Tang, K.; Lu, Y.; Tian, Z.; Huang, Z.; Wang, M.; Zhao, J.; Xie, J. Revealing the role of glycerophospholipid metabolism in asthma through plasma lipidomics. Clin Chim Acta2021, 513:34-42.
  6. Kennelly, J.P.; Carlin, S.; Ju, T.; van der Veen, J.N.; Nelson, R.C.; Buteau, J.; Thiesen, A.; Richard, C.; Willing, B.P.; Jacobs, R.L. Intestinal Phospholipid Disequilibrium Initiates an ER Stress Response That Drives Goblet Cell Necroptosis and Spontaneous Colitis in Mice. Cell Mol Gastroenterol Hepatol2021, 11(4):999-1021.
  7. Wang, Y.; Sun, X.; Qiu, J.; Zhou, A.; Xu, P.; Liu, Y.; Wu, H. A UHPLC-Q-TOF-MS-based serum and urine metabolomics approach reveals the mechanism of Gualou-Xiebai herb pair intervention against atherosclerosis process in ApoE-/-mice. J Chromatogr B Analyt Technol Biomed Life Sci2022, 1215:123567.
  8. Liu, P.; Zhu, W.; Chen, C.; Yan, B.; Zhu, L.; Chen, X.; Peng, C. The mechanisms of lysophosphatidylcholine in the development of diseases. Life Sci2020, 247:117443.
  9. Petkevicius, K.; Virtue, S.; Bidault, G.; Jenkins, B.; Çubuk, C.; Morgantini, C.; Aouadi, M.; Dopazo, J.; Serlie, M.J.; Koulman, A.; et al. Accelerated phosphatidylcholine turnover in macrophages promotes adipose tissue inflammation in obesity. Elife2019, 8: e47990.
  10. Zhou, X.; Wang, R.; Zhang, T.; Liu, F.; Zhang, W.; Wang, G.; Gu, G.; Han, Q.; Xu, D.; Yao, C.; et al. Identification of Lysophosphatidylcholines and Sphingolipids as Potential Biomarkers for Acute Aortic Dissection via Serum Metabolomics. Eur J Vasc Endovasc Surg2019, 57(3):434-441.
  11. Gonçalves, I.; Edsfeldt, A.; Ko, N.Y.; Grufman, H.; Berg, K.; Björkbacka, H.; Nitulescu, M.; Persson, A.; Nilsson, M.; Prehn, C.; et al. Evidence supporting a key role of Lp-PLA2-generated lysophosphatidylcholine in human atherosclerotic plaque inflammation. Arterioscler Thromb Vasc Biol2012, 32(6):1505-12.
  12. Alewijnse, A.E.; Peters, S.L. Sphingolipid signalling in the cardiovascular system: good, bad or both? Eur J Pharmacol2008, 585(2-3):292-302.
  13. Gault, C.R.; Obeid, L.M.; Hannun, Y.A. An overview of sphingolipid metabolism: from synthesis to breakdown. Adv Exp Med Biol2010, 688:1-23.
  14. Augé, N.; Maupas-Schwalm, F.; Elbaz, M.; Thiers, J.C.; Waysbort, A.; Itohara, S.; Krell, H.W.; Salvayre, R.; Nègre-Salvayre, A. Role for matrix metalloproteinase-2 in oxidized low-density lipoprotein-induced activation of the sphingomyelin/ceramide pathway and smooth muscle cell proliferation. Circulation2004, 110(5):571-8.
  15. Brown, E.M.; Ke, X.; Hitchcock, D.; Jeanfavre, S.; Avila-Pacheco, J.; Nakata, T.; Arthur, T.D.; Fornelos, N.; Heim, C.; Franzosa, E.A.; et al. Bacteroides-Derived Sphingolipids Are Critical for Maintaining Intestinal Homeostasis and Symbiosis.Cell Host Microbe2019, 25(5):668-680.e7.
  16. Yang, Z.H.; Nill, K.; Takechi-Haraya, Y.; Playford, M.P.; Nguyen, D.; Yu, Z.X.; Pryor, M.; Tang, J.; Rojulpote, K.V.; Mehta, N.N.; et al. Differential Effect of Dietary Supplementation with a Soybean Oil Enriched in Oleic Acid versus Linoleic Acid on Plasma Lipids and Atherosclerosis in LDLR-Deficient Mice.Int J Mol Sci2022, 23(15):8385.
  17. Yuan, X.; Nagamine, R.; Tanaka, Y.; Tsai, W.T.; Jiang, Z.; Takeyama, A.; Imaizumi, K.; Sato, M. The effects of dietary linoleic acid on reducing serum cholesterol and atherosclerosis development are nullified by a high-cholesterol diet in male and female apoE-deficient mice. Br J Nutr2022, 1-8.

Round 2

Reviewer 2 Report

Article is much improved. For future studies I would recommend sharing data through a public repository.